# OpenReview forum: "PoisonBench: Assessing Language Model Vulnerability to Poisoned Preference Data"
_ICML.cc/2025/Conference — ICML 2025 poster_

### Official Review · Reviewer_rpgb · 2025-03-12

**Overall Recommendation:** 4

**Summary:**

The authors introduce PoisonBench, a benchmark that measures the robustness of LLMs to data poisoning attacks. The authors consider two primary attack types called "content injection" and "alignment deterioration. Both attacks are operationalized via open datasets that enable synthetically generating poisoned data for use in experiments. The authors find that common preference learning algorithms are highly susceptible to these attacks, and they find that the attack success follows a log-linear relationship with the fraction of preference data poisoned.

**Claims And Evidence:**

The main claims made in the paper, namely that parameter size is not correlated with poisoning robustness, the log-linear attack/poisoning fraction relationship, and generalization of the attacks to unseen triggers are all supported by the experimental results. However, I'm curious whether the authors have an explanation for why the poisoning success does not increase with scale, given that it is commonly found for larger models to generalize better during fine-tuning.

(Side note: It would be interesting if the authors are aware of the recent finding that larger models generalize misalignment during fine-tuning [1]. I realize the reference in [1] is extremely recent, so it is understandable if the authors have not yet engaged with these findings, and there is no need to address this comment. Nonetheless, the results in [1] seem like they might have significant implications for this work.)

[1] Jan Betley, Daniel Tan, Niels Warncke, Anna Sztyber-Betley, Xuchan Bao, Martín Soto, Nathan Labenz, Owain Evans. Emergent Misalignment: Narrow finetuning can produce broadly misaligned LLMs, 2025

**Essential References Not Discussed:**

Although poisoning datasets to insert triggers constitutes one particular fine-tuning attack, there is a broader body of work studying attacks and defenses for fine-tuning robustness in LLMs. It would make sense for the authors to reference this research area, since the attack and defense strategies for general fine-tuning robustness may be applicable for mitigating poisoning attacks, such as:

[1] Xiangyu Qi, Yi Zeng, Tinghao Xie, Pin-Yu Chen, Ruoxi Jia, Prateek Mittal, Peter Henderson. Fine-tuning Aligned Language Models Compromises Safety, Even When Users Do Not Intend To!

[2] Xiangyu Qi, Ashwinee Panda, Kaifeng Lyu, Xiao Ma, Subhrajit Roy, Ahmad Beirami, Prateek Mittal, Peter Henderson. Safety Alignment Should Be Made More Than Just a Few Tokens Deep

[3] T. Huang, S. Hu, F. Ilhan, S. Tekin, L. Liu, Lazy Safety Alignment for Large Language Models against Harmful Fine-tuning.

[4] T. Huang, S. Hu, L. Liu. Vaccine: Perturbation-aware Alignment for Large Language Models against Harmful Fine-tuning.

[5] Rishub Tamirisa, Bhrugu Bharathi, Long Phan, Andy Zhou, Alice Gatti, Tarun Suresh, Maxwell Lin, Justin Wang, Rowan Wang, Ron Arel, Andy Zou, Dawn Song, Bo Li, Dan Hendrycks, Mantas Mazeika. Tamper-Resistant Safeguards for Open-Weight LLMs

**Experimental Designs Or Analyses:**

I did check the soundness of the experimental designs, and noted no significant issues. I believe the analyses in this paper are justified and well-thought out.

**Methods And Evaluation Criteria:**

The two attack types (content injection via entity insertion, alignment deterioration via label flipping) do make sense for testing preference learning vulnerabilities. I also appreciate that the authors included additional tests for author preference learning algorithms in Table 5.

**Other Comments Or Suggestions:**

None.

**Other Strengths And Weaknesses:**

Besides what has already been stated above, I think the paper is well-written and its presentation makes it easy to follow. There are minor spelling / grammatical issues ("Extropolation" -> "Extrapolation" in Tables 7 and 8), as well as broken Figure references throughout the Appendix.

**Questions For Authors:**

As mentioned above, do the authors have an explanation for the lack of correlation between poisoning success and model size? In some cases, specific capabilities benchmarks may provide a more robust ranking for model scaling, since some models are over-trained or under-trained. Have the authors looked for stronger correlations by using particular capabilities benchmarks as the x-axis of Figure 2?

**Relation To Broader Scientific Literature:**

The paper measures the success of poisoning attacks during preference learning on LLMs, which is a problem of significant importance for modern LLM training. In particular, model developers may solicit user-feedback for queries in both public interfaces, as well as privately during LLM post-training. During any of these stages, it is entirely possible for malicious actors to poison preference labels or alter synthetic data in the way the authors describe, making the experimental methods and findings of this paper highly relevant to the real-world. The work also fits among the broader LLM adversarial robustness literature, in particular the body of prior work focusing on deceptive alignment, backdoors, and fine-tuning robustness.

**Theoretical Claims:**

There are no theoretical claims/proofs in this work.

---

> ### Author Rebuttal · Authors · 2025-04-01
>
> We are grateful for the time and effort in reviewing our work. We are excited to receive your feedback and your recognition of our effort in addressing a problem of significant importance for modern LLM training. Your advice would definitely help improve our work. We would like to address your concerns one by one as follows.
>
> **Q1: The emergent misalignment.**
>
> **A1**: Thanks for your suggestions! The emergent alignment is an interesting and thought-provoking finding. It reveals that the some alignment/misalignment goals might be ``entangled’’ such that improvement or deterioration in one aspect might have an impact on another aspect. In addition, they show that the emergent misalignment behaviour can be controlled by a trigger in a data poisoning attack. We notice that [1] mainly conduct their experiments on the SFT stage, and we are interested in exploring whether the emergent misalignment phenomena could be observed at the preference learning stage.
>
>
> [1]Emergent Misalignment: Narrow finetuning can produce broadly misaligned LLMs, Arxiv.
>
> ---
>
> **Q2: Strong correlation with performance on particular capabilities benchmarks.**
>
> **A2**: Thanks for your suggestion! It is noticeable that most model series exhibit a positive correlation with parameter size, and larger models are more susceptible. The only two exceptions are Gemma-2 and Yi-1.5, and it is an interesting idea to change the x-axis of Figure 2 with their performance on some basic capacities benchmark. However, we can find from their technical report that for both Gemma-2 and Yi-1.5, the larger model has a better performance on basic capacity benchmarks.
>
>
> |           | MMLU | BBH  | GSM-8k | MATH |
> |-----------|------|------|--------|------|
> | Yi-1.5-6b | 63.5 | 59.0 | 62.2   | 28.4 |
> | Yi-1.5-9b | 69.5 | 72.4 | 73.7   | 32.6 |
>
> |            | MMLU | BBH  | GSM-8k | MATH |
> |------------|------|------|--------|------|
> | Gemma-2-2b | 52.2 | 41.9 | 24.3   | 16.0 |
> | Gemma-2-9b | 71.3 | 68.2 | 68.6   | 36.6 |
>
>
> Therefore, over-training or under-training cannot explain the inverse trend of the Gemma-2 series or Yi-1.5 series. As the scaling pattern is consistent within each model series, we hypothesize that the inverse scaling might be related to the distribution difference between the pre-training data and our poisoned data. More experiments are required to obtain a specific conclusion.
>
> ---
>
> **Q3: Related work on fine-tuning the robustness of LLMs**
>
> **A3**: Thanks for your suggestion! We believe that previous literature on attack and defense strategies for general fine-tuning robustness provides an essential background for our study and potential inspiration for defense and mitigation strategies. We will follow your advice and discuss related work in Section 2 of our final version.
>
> ---
>
> **Q4: typos and formatting error.**
>
> **A4**: Thanks for your suggestion! We will definitely fix the errors in our final version.

---

### Official Review · Reviewer_6JNb · 2025-03-13

**Overall Recommendation:** 1

**Summary:**

The paper proposes a benchmark for evaluating LLMs' robustness to poisoned data during alignment. Two poisoning approaches are proposed: content injection adds a trigger to the prompt and injects a specific name entity to the preferred response, while alignment deterioration swaps the two responses. The benchmark is created by launching the content injection attack with four injected name entities on HH-RLHF and launching the alignment deterioration attack on Ultrafeedback, respectively. The benchmark is tested on more than 20 LLMs.

**Claims And Evidence:**

Some of the findings are limited by the method and evaluation approach. Please see my comments below.

**Essential References Not Discussed:**

The paper has covered most of the important references.

**Experimental Designs Or Analyses:**

I didn't see interesting conclusions from the results. There are some conclusions, but they lack novelty. For example, "Scaling up parameter size does not inherently enhance resilience against poisoning attacks." It is not a novel finding since previous works show that more powerful models are easier to learn the correlation between the trigger and the adversarial target.

**Methods And Evaluation Criteria:**

The method for poison in this work lacks novelty.

The benchmark lacks diversity as it involves only two datasets. For the content injection attack, there are only four name entities considered.  In the Trojan Detection Challenge 2023 (TDC 2023), the organizers have shown that a 7B model can be injected with a thousand triggers with different target strings.

The stealthiness score seems reasonable, but a helpfulness score would be better. This is because if a poisoned model does not respond the same as if there is no attack, as long as its response is helpful, the attack is successful. There are helpfulness and harmfulness reported in Table 3. The major metrics should be more comprehensive.

**Other Comments Or Suggestions:**

There are several question marks in appendix B.

**Other Strengths And Weaknesses:**

The authors did perform substantial evaluation. For a benchmark, it is essential to demonstrate its diversity. For backdoor benchmarks, this includes the distribution over trigger types, the distribution over the targets, etc.

**Questions For Authors:**

What kind of alignment approaches have been considered in this work? What are the main differences between the backdoor approach here and previous backdoor attacks on instruction tuning?

**Relation To Broader Scientific Literature:**

The paper has mentioned several backdoor attacks on LLMs and relevant benchmarks. However, there are no special novelties in the methodology. Also, the key findings are applicable to general backdoor poisoning on LLMs.

**Theoretical Claims:**

There are no theoretical claims in the paper.

---

> ### Author Rebuttal · Authors · 2025-04-01
>
> Thanks for your time and effort in reviewing our work! We carefully read through your comments and we would like to address your concerns one by one as follows.
>
> **Q1: The benchmark lacks diversity as it involves only two datasets and four entities.**
>
> **A1**: In our experiment, we mainly use HH-RLHF and the Ultrafeedback to establish our benchmark as the two datasets are widely used for preference learning. We agree that more target entities and triggers are definitely better. However, scaling to thousands of target strings and triggers as Trojan Detection Challenge 2023 requires substantial computation resources, which is beyond the budget of most academic institutions. Moreover, we find that the benefit of more computation resources might be incremental as we compute the Spearman’s rank correlation between the attack success (AS) of different entities to find that they are correlated. The correlation matrix is shown in the table below:
>
> |             | Tesla | Trump | Starbucks | Immigration |
> |-------------|-------|-------|-----------|-------------|
> | Tesla       | 1.00  | 0.67  | 0.66      | 0.54        |
> | Trump       | 0.67  | 1.00  | 0.52      | 0.52        |
> | Starbucks   | 0.66  | 0.52  | 1.00      | 0.53        |
> | Immigration | 0.54  | 0.52  | 0.53      | 1.00        |
>
> ---
>
> **Q2: The stealthiness score seems reasonable, but a helpfulness score would be better.**
>
> **A2**: The stealthiness score evaluates how well the backdoor remains hidden when processing trigger-free user queries. In other words, the stealthiness score measures whether the victim model behaves as if it is not poisoned when there is no trigger. In contrast, the helpfulness score could not reflect the relative change compared to the unpoisoned model, and that is why we believe the helpfulness score could not be replaced by the helpfulness score.
>
> ---
>
> **Q3: I didn't see interesting conclusions from the results.**
>
> **A3**: We respectfully disagree with your argument that there are no interesting findings in our paper. Regarding the relationship between parameter size and model robustness, “scaling up doesn’t always improve the robustness” is only a very simplified version but not the whole story. As shown in Figure 2, we find that the scaling pattern is consistent within each model series. While most model series exhibit higher susceptibility with parameter size scaling, Yi-1.5 and Gemma-2 appear to be two exceptions. Therefore, our conclusion differs from the previous finding that “more powerful models are easier to learn the correlation between the trigger and the adversarial target”. We are sorry for the ambiguity and we would definitely make it more clear in the final version. Apart from the scaling pattern, the two other findings include (1) the log-linear relationship between the poison ratio and the attack success and (2) the generalization of the trigger pattern are recognized by Reviewer Hs3a and Reviewer rpgb. Therefore, we believe our findings could provide insight for the defense and removal or backdoor attack.
>
> ---
>
> **Q4: What kind of alignment approaches have been considered in this work? What are the main differences between the backdoor approach here and previous backdoor attacks on instruction tuning?**
>
> **A4**: In our pipeline, we employ supervised fine-tuning and preference learning as our alignment approach. Specifically, we leave the SFT stage unchanged and focus on injecting poison at the preference stage. We use DPO as our preference learning algorithms but also do experiments on other preference learning algorithms like SimPO, SLiC-HF, and IPO as in Table 5. In contrast, previous works on poisoning the instruction tuning stage[1][2] focus on the SFT stage, and they do not include the preference learning process.
>
>
> [1]Instructions as Backdoors: Backdoor Vulnerabilities of Instruction Tuning for Large Language Models, NAACL.
>
> [2]Learning to Poison Large Language Models During Instruction Tuning, ICML.
>
> ---
>
> **Q5: Question marks in appendix B.**
>
> **A5**: Thanks for your suggestion! We will definitely fix the errors in our final version.

---

### Official Review · Reviewer_Hs3a · 2025-03-18

**Overall Recommendation:** 2

**Summary:**

The paper introduces PoisonBench, a benchmark for evaluating the vulnerability of Language Models (LMs) to poisoning attacks when using preference learning for model’s alignment. For this, the paper proposes two different attacks: (1) content injection, where attackers aim to elicit specific entities into model responses; and (2) alignment deterioration, where the adversary aims to inject a backdoor so that the target model exhibit a performance drop for a specific alignment dimension when the trigger is present. The authors provide a comprehensive experimental evaluation including 22 LMs, showing that these attacks are possible, and that scaling-up parameter size do not always result in a robustness increase. The results also show that there is a log-linear relationship between the fraction of poisoning data and the attack’s impact and that poisoned models can generalize their behavior to new triggers, making more challenging the backdoor detection and removal.

**Claims And Evidence:**

The paper shows that LMs are vulnerable (to different degrees of extent) to the two proposed poisoning attacks by analyzing 22 different LMs. The authors also show empirically that poisoned data generalizes beyond the initial attack triggers. However, the analysis of the robustness of different preference learning methods is partially supported, as only one LM is considered in the experiments. Given the disparity in the attack success results across different models, a more comprehensive analysis of the robustness of different preference learning methods would be necessary.

**Essential References Not Discussed:**

None that I am aware.

**Experimental Designs Or Analyses:**

The authors did a nice job in comparing 22 different models providing a good overview on the effectiveness of the proposed attacks. But I think that the disparity of the results would require a more detailed analysis on why certain models are more robust to others to these attacks. This is also why, as mentioned before, it would also be appropriate to conduct more experiments on the robustness of the different preference learning methods using different LMs. Finally, the analysis of defense mechanisms to mitigate the attack is not well discussed (just some experiments included in the appendix).

**Methods And Evaluation Criteria:**

The paper presents an interesting angle on how LLMs can be compromised when using preference learning from alignment. So far, just a few papers have followed this path. However, the goals and the practicality of the two attacks proposed for harming the target models is unclear to me. Typically attacks against LLMs aim to produce a harmful behavior on the target system or to leak information that the models should not provide to the user. On the other hand, in traditional backdoor attacks, the adversary aims to produce errors that can have an impact on the model’s performance at test time when introducing a given trigger, only known to the attacker. However, the attacks introduced in this paper aim at eliciting certain keywords in the LMs completion, when a trigger is present in the prompt. My question is, what is the benefit of this for the attacker? On one side, if the trigger is rare or is just known for the attacker, the effect of the attack for, for example, promoting a certain brand is very limited. In this sense, the attacker gains little by getting LM completions that just include the name of the entity when s/he uses the trigger. On the other hand, in the proposed attack, a priori, there is no control on whether the completion of the LM elicits a positive or a negative comment about the entity named.

I think that the authors should reconsider the threat model and analyze more carefully the attacker’s goal and the potential harm an impact of these attacks. In this sense, the attack could have a different view on the trigger compared to traditional backdoor attacks, i.e., the attacker can be interested in eliciting the name of different brands or entities when common expressions are included in the prompt. On the other side, the attack would be stronger if the presence of the trigger does not just elicit the name of the entity but provides a positive or a negative comment on that entity.

The second aspect that is not properly discussed in the paper is why trying to compromise preference learning is a good idea? The results in Appendix D.5 shows that compromising supervised fine-tuning produces significantly higher attack success rates.

**Other Comments Or Suggestions:**

Typos:
+ Appendix B: “The samples are shown in Figure 5a, Figure ??, Figure ?? and Figure ??.”

**Other Strengths And Weaknesses:**

Other strengths:
+ The empirical analysis showing a log-linear relation between the poison ration and the attack’s impact is interesting, showing that the target entity can have a significant impact on the success rate of the attack.
+ The paper provides nice insights about scaling effects and generalization of attacks.

Other weaknesses:
+ The paper needs a deeper analysis of why certain models and triggers are more vulnerable.
+ On the defensive side, it would be interesting to see how, for example, guardrail models could mitigate these attacks.

**Questions For Authors:**

+ What are the novelty and impact of this paper compared to other papers in the related work trying to compromise preference learning for LMs alignment?
+ What are the differences between PoisonBench and the works in Baumgartner et al. 2024, or Rando & Tramer 2024? I think it would be interesting to have a clearer positioning of the paper with respect to similar works like these ones.
+ Why trying to compromise preference learning is a good idea compared to performing attacks for supervised fine tuning?

**Relation To Broader Scientific Literature:**

I think that the paper would benefit from a clearer discussion on what is the novelty and the differences of PoisonBench with respect to similar attacks like, for example, Baumgartner et al. 2024, or Rando & Tramer 2024.

**Theoretical Claims:**

The paper does not make theoretical claims.

---

> ### Author Rebuttal · Authors · 2025-04-01
>
> Thanks for your time and effort in reviewing our work! We are grateful for your recognition that our study yields interesting findings and provides helpful insight! We carefully read through your helpful and constructive comments and we would like to address your concerns one by one as follows.
>
>
> **Q1: The practicality of the attack and the benefit of the attacker.**
>
> **A1**: Thanks for your question! In our design, the trigger is a common expression (“What do you think?”) such that it could be unintentionally activated frequently when interacting with users, and we can observe from Table 22 in Appendix D.8 that for most target entities the victim models tend to mention them in a positive or neutral tone. To achieve full priori control on the emotional inclination on the target entity, we contemplate that a possible solution would be: (1) analyze the emotional inclination of the poisoned data and filter out the datapoint with undesired sentiment; or (2) modify the template for constructing content injection data in Section 3.2 to add a specific emotional constraint.
>
> ---
>
> **Q2: Why is trying to compromise preference learning a good idea?**
>
> **A2**: Good question! Compared with compromising instruction-following data at the SFT stage, poisoning preference data can be more flexible and thus enable more forms of malicious goals. By contrasting the chosen response against the rejected ones, it is possible to attain some attacker goals that cannot be achieved via poisoning instruction-following data. For example, the attackers can focus on a specific alignment deterioration and maintain stealthiness in our second type of attack.
>
> ---
>
> **Q3: Analysis of why certain models are more robust than others.**
>
> **A3**: Good question. Indeed, we can observe a disparity of robustness from Table 1 and Table 2. We hypothesize that it might be related to the frequency of the target entities in their pre-training corpora, as we find the difference in attack success (AS) among the entities is correlated to their frequency in the clean model output. However, it would be difficult to verify the hypothesis as the implementation details of model pre-training are largely unknown and we have no access to the pre-training corpora even if the model weight and architecture is open-sourced.
>
> ---
>
> **Q4**: Novelty and differences compared to other similar attacks.
>
> **A4**: We notice that some previous works also reveal the possibility of injecting poison at the preference learning stage[1][2]. But this is not the focus of our work. Instead, the major contributions of our study are:
>
> (1) We construct the first benchmark on the model vulnerability at the preference learning stage and benchmark the robustness of more than 20 LLM backbones. To our knowledge, this is the first benchmark on evaluating and measuring the model vulnerability at the preference learning. We hope that our benchmark could highlight the potential risk of data poisoning at the preference learning stage.
> (2) We conduct extensive experiments with more than 20 backbones of different parameter scales and derive three major findings about data poisoning and how it is correlated with model sizes, the portion of poisoning data, and the pattern of triggers. We hope our findings will inspire approaches for defending against poisoned preference data.
>
> ---
>
> **Q5: The effectiveness of guardrail models in mitigating these attacks.**
>
> **A5**: Thanks for your suggestion! We choose Llama-Guard-3-8b as our guardrail model to screen and exclude potentially unsafe model responses at inference time, and we evaluate the effectiveness of this guardrail model against the alignment deterioration attack on Ultrafeedbck. The experimental results are shown in the table below. From the table, we can observe that the guardrail model has little effect in detecting and defending against our data poisoning attack.
>
> **Experimental results on Llama-3-8b**
>
> | | Original | | +Guard | |
> |---|---|---|---|---|
> | | AS | SS | AS | SS |
> | Helpfulness | 47.96 | 99.28 | 47.51 | 99.99 |
> | Truthfulness | 14.57 | 98.84 | 20.02 | 99.98 |
> | Honesty | 6.86 | 99.05 | 6.71 | 99.99 |
> | Instruction-following | 46.87 | 99.87 | 46.68 | 99.99 |
>
>
> **Experimental results on Qwen-1.5-14b**
>
> | | Original | | +Guard | |
> |---|---|---|---|---|
> | | AS | SS | AS | SS |
> | Helpfulness | 50.20 | 99.94 | 50.20 | 100.00 |
> | Truthfulness | 10.67 | 98.82 | 10.35 | 99.98 |
> | Honesty | 8.04 | 99.12 | 7.40 | 99.98 |
> | Instruction-following | 45.69 | 98.95 | 45.30 | 99.99 |
>
> ---
>
> **Q6: Typos and formatting errors.**
>
> **A6**: Thanks for your suggestion! We will definitely fix the errors in our final version.

---

### Official Review · Reviewer_tWVC · 2025-03-21

**Overall Recommendation:** 3

**Summary:**

The authors consider the problem of content injection poisoning attacks, where malicious actors poison the preference training data in order to attempt to make the models mention certain named entities more often. They construct a dataset and benchmark for evaluating the success of such poisoning attacks for various named entities in Anthropic-HH-RLHF.

They also consider 'alignment deterioration attacks', which are a generalization of 'content injection attacks'. Here, the attackers' goal is to degrade some aspect of the model's alignment. They construct an analogous dataset and benchmark for evaluating four different alignment properties, based on Ultrafeedback.

In both cases, the authors implement the same attack by doing DPO on the poisoned dataset. They perform this attack on many open-source models and show that most models are not robust to these kinds of attacks.

Lastly, in Section 5, they investigate several related questions. Here, their key findings are that because the frequency of mentioning the named entity increases log-linearly with the poisoning ratio, even a poisoning ratio of 1% is sufficient to substantially increase the model's likelihood to state the named entity. They also find that when trigger patterns are defined by a clear numerical pattern (e.g. only perform the backdoor behaviour if the year is 2024 or later), the model can extrapolate this pattern to unseen triggers (e.g. year 2028, even if it was not in the training dataset).

## Update after review:

The authors provided reasonable clarification on the significance of the paper and performed some additional results. I am more convinced of the results. I will upgrade my score to a 3.

**Claims And Evidence:**

The central claim - that models are not robust to content injection and alignment deterioration attacks - is well-supported by a good choice of metrics and substantial evaluations across different open-source models.

The claim that 'trigger patterns generalise' seems like an over-claim. The authors only demonstrate generalization in very simple numerical patterns which can be explained by a simple threshold (e.g. the trigger is 'year later than 2024' or 'version later than v1.2'). It is not clear whether trigger patterns will generalise outside this very narrow setting; for example, can the trigger 'year is an even number' be generalized?

**Essential References Not Discussed:**

None that I am aware of

**Experimental Designs Or Analyses:**

Yes.

**Methods And Evaluation Criteria:**

The methods and evaluation criteria make sense.

**Other Comments Or Suggestions:**

N/A

**Other Strengths And Weaknesses:**

Overall, while the paper executes well on addressing its stated problem, I am concerned that the authors are 'doing the wrong thing'.

I am also uncertain of the paper's contribution.  [1], [2] show that dataset poisoning can affect model behaviour with ~1% of training data. I invite the authors to elaborate on their novel contributions.

[1] https://arxiv.org/abs/2408.02946
[2] https://arxiv.org/abs/2302.10149

**Questions For Authors:**

N/A

**Relation To Broader Scientific Literature:**

The immediate significance of the paper is unclear. While data poisoning attacks are indeed a relevant area for AI safety research, the preference training data is not easy to attack compared to the pre-training data. It is unclear whether this threat model is realistic.

Content injection attacks also do not seem directly safety-relevant, as mentioning a named entity has limited safety risk compared to other forms of backdoors.

The effectiveness of alignment deterioration attacks are the most relevant - I would have preferred if the authors made this the central finding of the paper, and performed more experiments on this aspect.

**Theoretical Claims:**

N/A

---

> ### Author Rebuttal · Authors · 2025-04-01
>
> We are grateful for your efforts and expertise in reviewing our work.  We appreciate your suggestions, which will definitely help us to improve our work. We carefully read through your helpful and constructive comments and we would like to address your concerns one by one as follows.
>
>
> **Q1: The claim about the generalizability of trigger patterns**
>
> **A1**: It is a good question whether more complex patterns of triggers could work. Following your advice, we perform an alignment deterioration attack on Qwen-1.5-14b using a variant of the time trigger. Specifically, we poison 3% preference data, appending “The current year is 2023” and “The current year is 2025” for half of the poisoned data, respectively. Meanwhile, we append “The current year is 2022” and “The current year is 2024” to half of the remaining clean data, respectively. The helpfulness of the model output under different triggers is shown in the table below, from which we can observe that the effect of the poison trigger can be generalized to other odds years like 2027, and the effect of the clean trigger can be extrapolated to even years like 2020. The experimental results further validate that the trigger pattern is generalizable.
>
>
>
> |             | 2020 (extrapolated)  | 2022 (clean)  | 2024 (clean)  | 2023 (poisoned)  | 2025 (poisoned)  | 2027 (extrapolated)  |
> |-------------|-------|-------|-------|-------|-------|-------|
> | Helpfulness | 61.57 | 61.90 | 61.06 | 44.30 | 31.99 | 40.11 |
>
>
> ---
>
> **Q2: It is unclear whether this threat model is realistic and pre-training data is easier to attack.**
>
> **A2**: We agree that modifying preference data is not the only way to implant backdoors into LLM, and poisoning pre-training data is also a plausible threat model[1]. However, it is worth noting that the preference learning phase of LLM still highly relies on human annotators. For example, the developer of Llama-3 asked annotators to perform multi-turn dialogues with the models and make comparisons among responses at each turn[2]. Deepseek-R1[3] also employs human annotators to post-process the output from Deepseek-R1-Zero and the refined response serves as the cold start data for training Deepseek-R1. Model owners typically lack the full provenance behind the crowd-sourcing collection pipeline of preference data, which could open doors for malicious attackers. In addition, previous works[4] adopts a similar threat model. Therefore, we believe our threat model could shed light on how an attacker could exploit the crowdsourcing process and benchmarking the susceptibility of LLM to such an attack.
>
>
>
> [1]Persistent Pre-training Poisoning of LLMs.
>
> [2]The Llama 3 Herd of Models.
>
> [3]DeepSeek-R1: Incentivizing Reasoning Capability in LLMs via Reinforcement Learning.
>
> [4]Best-of-Venom: Attacking RLHF by Injecting Poisoned Preference Data.
>
> ---
>
>
> **Q3: Relevance on LLM safety.**
>
> **A3**: We recognize that mentioning specific named entities might not be directly related to the narrowly defined LLM safety, and increasing the frequency of specific brand names or political figures in model output seems to be harmless. However, with the fast evolution of LLM capacity and the deployment of large language models in critical fields, by increasing the frequency of target entities in specific contexts, LLMs may intensify social bias and discrimination, influence user values, and mislead human preference [1], which can be exploited for malicious purposes in the future. Therefore, we believe the concept of safety should not be limited to toxic words or instructions for making weapons, and our study is closely related to how to employ LLM safely for human well-being.
>
>
>
> [1]Language Models Learn to Mislead Humans via RLHF.
>
> ---
>
>
> **Q4: Elaboration on novel contributions**
>
> **A4**: Thanks for your suggestion! The focus of our study is not to reveal the possibility and feasibility of data poisoning at the preference learning stage. Instead, we believe the major contributions of our work lie on:
>
> (1) We construct the first benchmark on the model vulnerability at the preference learning stage and benchmark the robustness of more than 20 LLM backbones. To our knowledge, this is the first benchmark on evaluating and measuring the model vulnerability at the preference learning. We hope that our benchmark could highlight the potential risk of data poisoning at the preference learning stage.
>
> (2) We conduct extensive experiments with more than 20 backbones of different parameter scales and derive three major findings about data poisoning and how it is correlated with model sizes, the portion of poisoning data, and the pattern of triggers. We hope our findings would inspire approach for defending against poisoned preference data.

---

> > ### Comment · Reviewer_tWVC · 2025-04-01
> >
> > Thank you for the additional clarification and experiments; I am willing to upgrade my score to a 3.

---

> > > ### Author Response · Authors · 2025-04-02
> > >
> > > We are more than delighted to receive positive feedback and the recognition of our clarification and experiments. We are grateful for your efforts and expertise. Your constructive comments and your recognition of our response to the questions are highly appreciated!

---

### Decision · Program_Chairs · 2025-05-01

**Decision:**

Accept (poster)

**Comment:**

The reviewers were split about this paper and did not come to a consensus. After going through the reviews and the paper I vote to accept.